# Comparative Genomics Reveals Novel Target Genes towards Specific Control of Plant-Parasitic Nematodes

**DOI:** 10.3390/genes11111347

**Published:** 2020-11-13

**Authors:** Priscila Grynberg, Roberto Coiti Togawa, Leticia Dias de Freitas, Jose Dijair Antonino, Corinne Rancurel, Marcos Mota do Carmo Costa, Maria Fatima Grossi-de-Sa, Robert N. G. Miller, Ana Cristina Miranda Brasileiro, Patricia Messenberg Guimaraes, Etienne G. J. Danchin

**Affiliations:** 1Embrapa Genetic Resources and Biotechnology, Brasília DF 70770-917, Brazil; priscila.grynberg@embrapa.br (P.G.); roberto.togawa@embrapa.br (R.C.T.); bioantonino@gmail.com (J.D.A.); marcos.costa@embrapa.br (M.M.d.C.C.); fatima.grossi@embrapa.br (M.F.G.-d.S.); ana.brasileiro@embrapa.br (A.C.M.B.); patricia.guimaraes@embrapa.br (P.M.G.); 2National Institute of Science and Technology—INCT PlantStress Biotech—EMBRAPA, Brasília DF 70770-917, Brazil; robertmiller@unb.br; 3Institute of Biological Sciences, Campus Universitário Darcy Ribeiro, University of Brasília, Brasília DF 70910-900, Brazil; leticiadfreitas.agro@gmail.com; 4Department of Agronomy-Entomology, Rural Federal University of Pernambuco, Recife PE 52171-900, Brazil; 5INRAE, Plant Health and Environment, University Côte d’Azur, CNRS, ISA, F-06903 Sophia-Antipolis CEDEX, France; corinne.rancurel@inrae.fr; 6Genomic Sciences and Biotechnology PPG, Catholic University of Brasília, Brasilia DF 71966-700, Brazil

**Keywords:** comparative genomics, plant-parasitic nematodes, phylogenomics, parasite-specific genes, pest control, *de novo* gene birth, horizontal gene transfers

## Abstract

Plant-parasitic nematodes cause extensive annual yield losses to worldwide agricultural production. Most cultivated plants have no known resistance against nematodes and the few bearing a resistance gene can be overcome by certain species. Chemical methods that have been deployed to control nematodes have largely been banned from use due to their poor specificity and high toxicity. Hence, there is an urgent need for the development of cleaner and more specific control methods. Recent advances in nematode genomics, including in phytoparasitic species, provide an unprecedented opportunity to identify genes and functions specific to these pests. Using phylogenomics, we compared 61 nematode genomes, including 16 for plant-parasitic species and identified more than 24,000 protein families specific to these parasites. In the genome of *Meloidogyne incognita*, one of the most devastating plant parasites, we found ca. 10,000 proteins with orthologs restricted only to phytoparasitic species and no further homology in protein databases. Among these phytoparasite-specific proteins, ca. 1000 shared the same properties as known secreted effectors involved in essential parasitic functions. Of these, 68 were novel and showed strong expression during the endophytic phase of the nematode life cycle, based on both RNA-seq and RT-qPCR analyses. Besides effector candidates, transcription-related and neuro-perception functions were enriched in phytoparasite-specific proteins, revealing interesting targets for nematode control methods. This phylogenomics analysis constitutes a unique resource for the further understanding of the genetic basis of nematode adaptation to phytoparasitism and for the development of more efficient control methods.

## 1. Introduction

Plant parasitism has emerged at least four times independently in the phylum Nematoda and encompasses approximately 15% of the whole nematode biodiversity [1]. Collectively, phytoparasitic nematodes cause considerable damage to worldwide agriculture, although only a few species, mainly in the clade Tylenchida, are responsible for the majority of the losses. Root-knot nematodes (RKN) of the genus *Meloidogyne* are the most devastating plant-parasitic nematodes, with susceptible hosts covering more than 4000 plant species, including the major economically important crops [2,3]. Among RKN, the tropical polyphagous apomictic species *M. incognita*, *M. javanica* and *M. arenaria*, together with the facultative sexual species *M. hapla* from temperate regions, are considered the most damaging. Annual global crop production losses due to root infection by these sedentary endoparasites exceed 5%, equating to more than US$ 80 billion in lost revenue [4,5]. Most severe yield losses occur in tropical regions, with the RKN wide host range restricting options for disease control through crop rotation. Within RKN species, this wide host range is subdivided into ‘host races’ with distinct ranges of host compatibilities. Recent population genomics analyses showed that this division in ‘host races’ is not phylogenetically supported and most likely resulted from recurrent and independent adaptations [6]. Thus, the term “host race” is probably not well-founded and the durability of crop rotation strategies is questionable. The employment of resistant cultivars to reduce the losses can also be limited due to the occurrence of ‘resistance-breaking’ virulent RKN isolates and species mixtures [7]. Furthermore, given their low specificity and potential risk for the environment and human health, chemical nematicides have now been mostly banned from use [8]. Hence, RKN control options are limited and advances in the development of novel sustainable and more specific control strategies are urgently required.

In RKN species, the infective second-stage juveniles (J2) penetrate host roots and migrate intracellularly to form specialized feeding cells that support the parasite throughout the sedentary life cycle. These different steps occur following the injection of secreted effector proteins produced in the nematode esophageal glands [3]. In the absence of host genes conferring resistance, nematode effectors will induce host cell differentiation into giant cells, which undergo nuclear enlargement, organelle proliferation, metabolic activation, cell-cycle alterations, and cell-wall changes [9]. This continued delivery of effector proteins modulates suppression of plant defenses and the full development of the giant cell, mediating the feeding process and the resultant metabolic drain on the host [10].

Effectors of RKN and other plant-parasitic nematodes (PPN) target important host molecular components to facilitate parasitism, being involved in the degradation of the plant cell wall, uptake, and processing of plant nutrients, manipulation of the plant defense system, or in the establishment of feeding structures [11,12]. The evolutionary origin of nematode effectors has so far been only poorly investigated. Nonetheless, previous studies showed that certain effector proteins, including those involved in the degradation of the plant cell wall (e.g., cellulases, pectinases), are conserved in many phytoparasitic nematodes and seem to have been acquired via horizontal gene transfer (HGT) from bacteria and fungi [13,14]. Still, the majority of known and candidate effectors represent so-called ‘orphan’ proteins, lacking any recognizable homology in species other than the PPN [11]. Even within PPN, comparison of the sets of known effectors in RKN and the cyst nematode *Globodera pallida*, showed only little overlap [15]. Although this could simply be due to a lack of comprehensive identification of nematode effector sets, this could also reveal lineage-specific effectors. Despite lineage-specific genes and those acquired via HGT, some effectors can also emerge from evolutionarily conserved genes, and acquire modifications in transcriptional regulation or protein localization signals to eventually be co-opted for new parasitic-related functions [16]. Several effectors form expanded multigene families, probably as a result of positive selective pressure imposed by plant hosts [12,13].

Given their role in parasitism, one compelling method towards combating PPN consists of silencing the expression of effector genes [17,18,19]. Describing more thoroughly the effector protein repertoire and dissecting their functions is therefore important for the development of novel nematode control strategies. Over the years, the availability of several RKN genomes provided new resources for the identification of complete sets of candidate effectors in these species [20,21,22,23,24,25,26,27,28,29]. Although nematode effectors are very diverse, many share common features with other secreted animal proteins, such as the presence of N-terminal signal peptides for secretion, or the absence of transmembrane domains. These two features can be used as signatures for bioinformatics identification of putative secreted proteins (PSP), but further filtering of effectors from the rest of the PSP requires consideration of other characteristics (e.g., gene expression patterns). Furthermore, correct identification of signal peptides for secretion can be hampered by inaccuracy of gene predictions at their 5′ ends in genomes, with certain known effector proteins also not bearing recognizable signal peptides. For these reasons, complementary approaches have compared known effectors to negative sets of presumably non-effector proteins in order to identify either an ensemble of characteristics specific to cyst nematode effectors via machine learning [30] or degenerate protein motifs specific to RKN effectors [31].

Predictions of candidate effectors in the first available RKN genomes (*M. incognita* and *M. hapla*) were initially conducted using the presence of a signal peptide for secretion of an RKN-specific motif and the absence of transmembrane domain as search filters [32]. Cross-referencing of this list with sets of proteins specific to RKN that were identified through a comparative genomics approach led to the identification of 15 novel candidate effectors without homologs in other species [32]. RNA interference (RNAi) experiments subsequently showed a significant reduction of nematode infection symptoms in tomato host plants for 12 of these novel candidate effectors. Being specific to phytoparasites, while displaying the same characteristics to known effectors, and yielding reduced nematode infection symptoms following silencing, these genes constitute promising novel targets for the development of more specific and environmentally friendly control methods. Searching for genes specific to phytoparasites constituted an original strategy in comparison to the other studies, which rather started from PPN genes orthologs to *Caenorhabditis elegans* genes known to yield severe phenotypes when inactivated [33]. Although this latter strategy maximizes the chance of severely affecting the PPN, their conservation across distantly related nematodes increases the risk of unintentional effects on the local nematofauna and possibly even beyond. 

Even if original, the comparative genomics study published in 2013 [32] was limited to the only two genomes for plant-parasitic species available at the time (two RKN). Since this initial comparative genomics study, the availability of genome sequences for additional RKN species [24,26,27,28,29], as well as for other plant-parasitic nematode species, including cyst nematodes [15,34,35], stem and bulb nematodes [36,37], or the more distantly related pine wilt disease nematode [38], has now greatly expanded the diversity of genomes for plant-parasitic nematode species. Similar genome sequencing efforts were also accomplished in the other nematode clades, including in animal parasites and free-living species, offering a novel opportunity for nematode comparative genomics. Recently, a massive comparative genomics analysis of nematodes, centered on human and other animal parasites, revealed the nematode-wide patterns of gene conservation, birth, and loss, and allowed for establishment of a prioritized list of new potential drug targets to combat these parasitic worms [39].

In the present study, we compared the genomes of 60 nematode species, carefully selected on the basis of completeness and quality of their predicted protein sets, in order to identify proteins exclusive to PPN. Our study encompassed 16 PPN species and represents an unprecedented insight into phytoparasite genomics in comparison with previous comparative studies on only two [32] or three [39] PPN genomes. We selected the genome of *M. incognita*, arguably one of the most damaging PPNs, to further identify proteins with the same characteristics as known effectors. Thus, they may also be involved in plant parasitism. We employed a bioinformatics functional annotation and enrichment analysis to assess whether certain functional categories were significantly enriched in the set of PPN-specific genes and in candidate effectors. Cross-referencing the list of candidate *M. incognita* effectors with PPN-specific proteins, we aimed to identify novel RKN proteins amenable to the development of specific and sustainable control methods. Since *in-planta* RNAi-based strategies constitute a promising approach, we then further analyzed life stage-specific RNA-seq data in *M. incognita* to identify candidate effector genes expressed in the endophytic phases of the life cycle, and employed qRT-PCR to confirm the expression patterns and validate them as promising targets. This comparative analysis constitutes a valuable resource for identification of new drug targets and for better understanding how PPN manipulate their hosts.

## 2. Materials and Methods 

### 2.1. Orthogroups Inference

A total of 2,228,060 proteins were downloaded from 79 nematode genomes and two tardigrade outgroup species. We assessed proteome completeness using BUSCOv3, according to the Eukaryotic odb9 dataset [40], setting up a minimal threshold of 70% complete BUSCO genes for inclusion in further analysis. For each species, when multiple proteome versions were available, only those with the highest BUSCO completeness scores were retained. This resulted in a selection of 63 protein sets covering 1,607,105 proteins, from nematodes with free-living, plant, vertebrate and insect parasite lifestyles, as well as two outgroup free-living tardigrade species. We employed OrthoFinder (version 2.3.3) with default inflation parameters [41,42] to define orthogroups between the 63 different species, based on an all-against-all comparison of the proteomes with Diamond [43] run in sensitive mode. The list of analyzed species, with BUSCO completeness scores, sequences per species, nematode lifestyles, as well as source links for data download and references, is available at https://doi.org/10.15454/IIAQOW.

We used the OrthoFinder option -m to generate multiple sequence alignments for each orthogroup using MAFFT [44], with phylogenetic trees generated with FastTree [45]. A species tree was inferred from a concatenated proteins tree using STAG, with the tree rooted using STRIDE [42]. The rooted species tree is available at https://doi.org/10.15454/MBMOQC. Global statistics of the OrthoFinder analysis and complete results are available at https://doi.org/10.15454/ZGUP7N and https://doi.org/10.15454/ZAYJBC, respectively.

### 2.2. Identification of M. incognita Putative Secreted Proteins and Candidate Effectors

Effector proteins are secreted by the nematode into the plant. In order to identify putative secreted proteins (PSP) in the *M. incognita* protein set, we searched for signal peptides as evidence for secretion using SignalP v4 [46]. This program improves the differentiation between transmembrane regions present in the N-terminal part of a protein and the presence of a signal peptide. To further circumvent this potential problem, predicted signal peptides were cropped, with the remaining part of the proteins subjected to TMHMM [47] for prediction of transmembrane regions. All proteins bearing a predicted signal peptide and no transmembrane region were considered as putative secreted proteins (PSP). Accession numbers of the proteins satisfying these two criteria are available at https://doi.org/10.15454/JCYZDI.

Pearson’s Chi-squared test with Yates’ continuity correction was applied to test whether the proportion of PSP was significantly different in PPN-specific predicted proteins vs. the whole predicted proteome. To further refine this set and identify candidate effectors, we searched for the presence of motifs enriched in known effectors in *M. incognita* [31]. We employed the MERCI software [31] to scan the whole set of *M. incognita* predicted proteins for the presence of at least one of the four known enriched motifs in the 100 first amino acids (list available at https://doi.org/10.15454/LMY6LV).

### 2.3. Taxonomic Distribution of M. incognita Homologs and Detection of Horizontal Gene Transfers

All *M. incognita* predicted proteins were compared against the NCBI’s nr library using Diamond [43], with an E-value threshold of 0.01. In order to explore the taxonomic distribution of nr hits matching *M. incognita* proteins, we employed the taxonomic assignment algorithm of Diamond, based on inference of the last common ancestor (LCA) of hits returning a Diamond score not diverging by more than 10% from the score of the best hit.

We employed Alienness [48] to detect putative horizontal gene transfers (HGT). Alienness parsed Diamond homology search results and retrieved taxonomic information associated with hits. The best metazoan and best non-metazoan hits were then identified to compute an Alien Index (AI), with self-hits to Tylenchomorpha (NCBI:txid33283) ignored. Proteins returning an AI > 0 have higher similarity to non-metazoan than metazoan hits in nr and can include candidate HGT of non-metazoan origin. Proteins having an AI > 0 and ≥70% identity with a non-metazoan hit were considered putative contaminants and eliminated from the list of candidate HGT. As AI ≥ 14 represents a good balance between precision and recall of phylogenetically confirmed HGT [48], we considered *M. incognita* proteins returning an AI ≥ 14 as likely to result from acquisition via HGT.

We then employed Pearson’s Chi-squared test with Yates’ continuity correction to test whether the proportion of putative HGT (AI > 0 and AI ≥ 14) were significantly different in PPN-specific predicted proteins vs. the whole predicted proteome. Finally, we identified enriched GO terms associated with candidate HGTs, using the same methodology explained above. All Alienness analysis results are available at https://doi.org/10.15454/W6SWZH.

### 2.4. Functional Annotation and Gene Ontology Terms Enrichment

All the predicted proteins from the *M. incognita* genome [23] were analyzed by InterProScan (v5.29–68.0) [49] for identification of conserved protein domains, with the option -iprlookup -goterms employed to assign Gene Ontology (GO) terms from the identified InterPro domains. Raw results of InterProScan annotation are available at https://doi.org/10.15454/9BFFKG.

We used the hypergeometric test within the FUNC program [50] to determine whether certain GO terms assigned via InterProScan annotation were significantly enriched in several subsets of *M. incognita* proteins. We made two independent analyses: (i) PPN-specific proteins—(a) PPN-specific proteins, (b) PPN-specific PSP and (c) PPN-specific candidate effectors and (ii) Proteins originating from HGT—(a) Possible HGT (AI > 0) and (b) Likely HGT (AI ≥ 14).

We considered as significantly enriched those terms returning a false discovery rate (FDR) lower than 0.05 and a minimum of four genes in node. When overlaps between proteins associated with several enriched GO were identified, these proteins were joined together into representative groups. Each group was given a more general term that represents and/or explains the biological processes and/or molecular functions in which they participate. Poorly informative general terms (higher in the hierarchy, usually inferred by FUNC) were further excluded from the analysis when the majority of the corresponding proteins in a node were spread out across more specific GO terms. GO terms fold-enrichment (FE) values were calculated as the ratio between the observed and the expected genes frequency. The R packages ggplot2 [51] and pheatmap were used to plot a bubble chart and heatmaps of enriched GO terms. Rows were colored based on the GO terms and the above-mentioned subsets of *M. incognita*.

We also used OrthoFinder results to assess whether *M. incognita* proteins corresponding to enriched GO terms were further conserved between RKN and other PPN (e.g., cyst, burrowing, and migratory plant-parasitic nematodes). As certain genomes are polyploids, the raw numbers of proteins in each orthogroup were not taken into account for this analysis. Results were plotted into a heatmap using R’s pheatmap package.

### 2.5. In-Silico Expression Patterns of Protein-Coding Genes throughout M. incognita Life Stages

In order to evaluate expression patterns of protein-coding genes, we analyzed data from a previously published life stage-specific RNA-seq analysis of *M. incognita* during infection in tomato [23]. This analysis encompassed four different life stages sequenced in triplicate: (i) eggs, (ii) pre-parasitic second stage juveniles (J2), (iii) a mix of late parasitic J2, third stage (J3) and fourth stage (J4) juveniles, and (iv) adult females. Stages (i) and (ii) correspond to exophytic phases while (iii) and (iv) are endophytic phases. The original cleaned RNA-seq reads [23] were retrieved and re-mapped to the *M. incognita* annotated genome assembly, using a more stringent end-to-end option (i.e., no soft clipping) in 2-passes as explained in [52]. Expression data, as FPKM values, are available at https://doi.org/10.15454/YM2DHE.

### 2.6. Candidate Effector Gene Expression Validation

#### 2.6.1. Nematode Population and Plant Inoculation

A population of *M. incognita* race 3, previously identified using esterase and SCAR markers [53], was maintained on tomato (*Solanum lycopersicum* ‘Santa Cruz’) in greenhouse conditions. *M. incognita* eggs were extracted from 60-day-old infected roots according to ref [54], hatched in vitro and pre-parasitic J2 concentrations adjusted to 500/mL. 

Bioassays were performed in triplicates, with 20-day-old tobacco plants (*Nicotiana tabacum*) inoculated with 1000 pre-parasitic J2, maintained at 25 °C, 12/12 h light/dark regime, and watered at 3-day intervals. Plants were harvested at 5, 10, 15, and 22 days after nematode inoculation (DAI). Roots were dissected under a binocular microscope to identify and collect RKN galls. Galls were immediately frozen in liquid nitrogen and maintained at −80 °C for RNA extraction. 

#### 2.6.2. Total RNA Extraction and cDNA Synthesis

Total RNA from eggs and pre-parasitic J2 were extracted using a DirectZol extraction kit (Zymo Research, Irvine, CA, USA), while for RNA extraction from *N. tabacum* root gall tissues at 5, 10, 15, and 22 DAI, we used the Quick-RNA^TM^ Plant Miniprep (Zymo Research, USA), according to each manufacturer’s standard protocol. After RNA extraction, samples were DNase I treated and cDNA synthesized using SuperScript II (Invitrogen, Carlsbad, CA, USA) and 1 µg of total RNA from each sample.

#### 2.6.3. RT-qPCR Analysis of Expression Patterns

The relative expression patterns of seven genes encoding new candidate effectors, throughout the nematode infective life cycle, were determined by reverse transcription quantitative PCR (RT-qPCR) assays. Each reaction was performed in a final volume of 10 µL, containing 5 µL of iTaq^TM^ Universal SYBR^®^ Green qPCR mix (Bio-Rad, Hercules, CA, USA), 0.2 µM of each primer and 2 µL of cDNA (diluted 1:20). PCR amplifications were conducted on the StepOne Real-Time PCR Systems (Applied Biosystems, Foster City, CA, USA) using the following program: 52 °C for 2 min, 95 °C for 10 min, followed by 40 cycles of 95 °C for 15 s and 60 °C for 60 s. Primers were designed using the Primer3Plus software [55] using the default qPCR parameters (Table 1). Primer specificity was assessed by analyzing the dissociation curve of amplified products using SDS software, version 2.2.2 (Applied Biosystems) and primer efficiencies were calculated using the online real-time PCR Miner tool [56]. The relative expression (RQ) of mRNA levels of target genes was calculated in samples using two reference genes (β-actin and 18S ribosomal subunit) by the SATqPCR web tool [57]. Experiments were conducted in biological and technical triplicate and the RQ was statistically tested by the analysis of variance (ANOVA) and Tukey’s test (*p* < 0.05).

## 3. Results

### 3.1. The Comparative Genomics Framework Covers a Diversity of Nematode Clades and Lifestyles

A total of 63 proteomes predicted from 61 nematode and two outgroup tardigrade genomes passed the BUSCO eukaryota completeness inclusion threshold of 70% (annotated species list available at https://doi.org/10.15454/IIAQOW).

The 61 nematode genomes covered species in four of the five nematode clades (I, III-V) defined by Mark Blaxter and colleagues [58,59] and seven of the twelve nematode clades (2, 6, 8–12) defined by Hans Helder and colleagues [1,60], respectively referred to hereafter as B-clades and H-clades. To date, no genome sequences are publicly available for nematodes either in Enoplia (H-clade 1, B-clade II) or in H-clades 3–5 and 7.

In terms of lifestyle, the selected nematode genomes encompassed nine free-living species, distributed in H-clades 6, 9, 10b and 11, three insect parasites, in H-clades 2 and 10a, 32 vertebrate parasites, in H-clades 2, 8, 9 and 10b, as well as 16 plant parasites, in H-clades 10d and 12 (Figure 1). Although plant parasitism has emerged at least four times independently in the phylum Nematoda, in H-clades 1, 2, 10 and 12 [1,61], genomes for PPN are currently only available in H-clades 10d and 12. While H-clade 10d is only represented here by the pine wilt disease agent *Bursaphelenchus xylophilus*, PPN in H-clade 12 now benefit from a much larger sampling, with genomes characterized for sedentary endoparasitic root-knot nematodes (*Meloidogyne* spp.) and cyst nematodes (*Heterodera* spp. and *Globodera* spp.), semi-endoparasitic reniform nematodes (*Rotylenchulus* spp.) and migratory endoparasitic nematodes (*Radopholus* spp and *Ditylenchus* spp.).

### 3.2. The Phylogenomics Analysis Is Consistent with the Nematode Phylogeny

The species tree produced by OrthoFinder (Figure 1) was almost identical to the most recent phylogenomics analysis based on concatenated evolutionarily conserved protein-coding genes [62] and consistent with the nematode classifications described by Mark Blaxter et al. [58,59] and Hans Helder et al. [1,60]. As expected, the two tardigrades were positioned as a monophyletic outgroup to the nematodes (Figure 1). Monophyly of all the nematode B-clades covered by our analysis (I, III, IV, V) was respected with strong STRIDE support values (all branch support values = 1). Similarly, all the H-clades covered were monophyletic and supported by the same strong values, with the exception of H-clade 10, which was split into 4 subclades, exactly as in [62]. In addition to the clades monophyly, their relative branching orders were also consistent with the nematode reference phylogeny and classification.

Considering that the Tylenchida group (H-clade 12) contains the most damaging PPN, we analyzed this clade in detail. In this group, the Anguinidae (*Ditylenchus destructor* and *D. dipsaci*) hold the earliest branching position, in accordance with previous phylogenetic analyses [1,60,62]. A clear separation was also observed between the Meloidogynidae (RKN) and a group composed of the Heteroderidae (cyst nematodes) and their closest available relatives, namely the Rotylenchulidae (*Rotylenchulus reniformis*) and the Radopholinae (*Radopholus similis*). Within the Meloidogynidae, a correct separation was observed between the three different RKN clades (I, II, III), as originally defined by Tandingan De Ley [63], and the branching order of these clades was respected.

Overall, the OrthoFinder-based tree topology received very high branch-level support values throughout, with most being higher than 0.9 (Figure 1). Only the relative positions of RKN within clade I displayed lower values, although the monophyly of this clade was itself supported by a value of 1.0 (see Section 4.3 for an explanation).

The overall high support of this phylogenomics-based tree topology, which is consistent with reference classifications, indicates that the phylogenetic signal present in our comparative genomics analysis allowed an accurate reconstruction of the nematode tree of life, therefore enabling a valid subsequent evolutionarily analysis.

### 3.3. Orthogroups Encompass the Vast Majority of Nematode Proteins and Reveal Genes Specific to Plant Parasites

More than 82.8% of the proteins from the 61 nematode and 2 tardigrade genomes (1,330,958/1,607,105 in total) were classified into 71,576 orthogroups by OrthoFinder (full results available at https://doi.org/10.15454/ZAYJBC). The median and average proportions of proteins assigned to orthogroups per species were 86.2 and 84.4%, respectively (full statistics available at https://doi.org/10.15454/ZGUP7N), therefore encompassing most of the diversity of proteins present across the different species selected for this analysis.

Overall, 289,165 proteins (<18% of the total) were species-specific and divided into two categories: (i) the great majority (276,147 proteins) could not be assigned to any orthogroup and represented species-specific singletons, while (ii) 13,018 proteins were assigned to single-species orthogroups and represent species-specific multicopy proteins. However, the proportion of species-specific proteins is not homogeneously distributed across the phylogenomics tree, with species being the sole representatives for their clades showing higher proportions of species-specific genes than species from well-represented clades. For instance, with 68% of its proteins being species-specific, the most extreme case is *Romanomermis culicivorax*, the sole representative of H-clade 2B (Figure 1). The other species from poorly represented branches also tend to show higher proportions of species-specific proteins. For example, *Plectus sambesi*, the only representative of H-clade 6, contains 29% of species-specific proteins. Similarly, the two *Steinernema* species, as sole representatives of H-clade 10a, both possess >35% of species-specific proteins. In contrast, species from well-represented clades showed a very low proportion of species-specific proteins. For instance, with seven species from the RKN genus present, *M. incognita* has only 2.5% of species-specific proteins. Similarly, *Elaeophora elaphi*, from H-clade 8, which is represented by 9 species, showed only 2.1% of species-specific proteins.

As PPN species are the main focal point of our study, we explored these data more closely. We identified 24,397 orthogroups (covering 155,876 proteins) that were exclusive to PPN. Only five of these orthogroups (covering 527 proteins) were universal to all the PPN present in this analysis, including *B. xylophilus*, the only PPN not belonging to H-clade 12 (Tylenchida). Excluding *B. xylophilus*, we found 27 PPN-specific orthogroups (covering 1750 proteins) that were universal to all the Tylenchida species genomes included here. Being specific and widely conserved in PPN, these proteins are likely to cover core functions, possibly including those linked to plant parasitism.

In the RKN *M. incognita*, we identified 12,683 proteins that were specific to PPN species, with more than 91% (11,620) distributed in 6526 PPN-specific orthogroups and only 1063 being *M. incognita*-specific singleton proteins (https://doi.org/10.15454/I9MWRS).

### 3.4. A Large Set of M. incognita Proteins Is Specific to PPN

According to the OrthoFinder results, approximately 12,700 *M. incognita* proteins had no ortholog in species other than PPN. However, this only holds true for the set of nematode and outgroup tardigrade species included in this comparative analysis. To further assess whether these proteins actually are orphans, without recognizable homology in other species beyond the nematode and tardigrades, we compared the whole set of *M. incognita* predicted proteins against the NCBI’s nr library.

In the whole *M. incognita* predicted proteome, more than 72% (31, 811/43, 718) of the entries returned at least one hit at an E-value threshold of 0.01. Using the last common ancestor algorithm included in Diamond, a total of 330 different taxonomic identifiers (taxids) could be assigned from the hits. As expected, 68% of the hits were assigned nematode species taxids, while metazoa as a whole represented 90% of taxid assignments. It should be noted that only 1403 hits (<5%) were assigned to the Meloidogyne genus or more broadly to the Tylenchina clade, reflecting the poor representation of plant-parasitic nematodes in the NCBI’s nr databank. Complementary approaches such as the nematode-centered orthology search that we carried out are thus necessary. Overall, eukaryota represented 94% of the hits in taxonomic annotations, while unassigned cellular organisms and bacteria represented only 4% and 0.7%, respectively, and might correspond to horizontal gene transfers or contaminants in the genome of *M. incognita*. Conversely, 11,907 proteins (>27% of the predicted proteins) did not return any hit against nr and might represent orphan proteins. The taxonomic distribution of *M. incognita* hits against nr, and the corresponding Krona graphs are available at https://doi.org/10.15454/FROF42.

By comparing the 12,683 *M. incognita* PPN-specific proteins identified by OrthoFinder and the Diamond homology search, we found that only 22% (2770) returned hits against the NCBI’s nr databank. In total, only 222 taxids could be assigned, and, in contrast to the whole proteome analysis, the phylum Nematoda was now a minority and represented only 41% of the taxonomic assignments (of which 27% were Tylenchida). Overall, more than 78% (9913/12,683) of the PPN-specific proteins identified by OrthoFinder returned no significant hit at all against the NCBI’s nr database. These proteins lack homology in other nematodes and more broadly in other species. The accession numbers of *M. incognita* proteins specific to PPN and returning no significant hit against the NCBI’s nr are available at https://doi.org/10.15454/48436V. 

### 3.5. Putative Secreted Proteins and Candidate Effectors Are Enriched in PPN-Specific M. incognita Proteins

We identified 2811 putative secreted proteins (PSP) in the genome of *M. incognita*, based on the presence of a predicted N-terminal signal peptide for secretion and an absence of predicted transmembrane region. These PSP are enriched in the PPN-specific *M. incognita* proteome, with almost half of them (1331/2811) falling in this category, while PPN-specific proteins represent less than one third of the *M. incognita* proteins (Pearson’s Chi-squared test with Yates’ continuity correction = 489.63, df = 1, *p*-value < 2.2 × 10^−16^). The complete list of *M. incognita* PSP and those specific to PPN are available at https://doi.org/10.15454/JCYZDI.

Effector proteins are secreted by nematodes in plant tissues and support parasitism through degradation of the plant cell wall, manipulation of plant defense and cell cycle, or through other mechanisms [12]. Although PSP will certainly include candidate effectors, protein motifs suggesting secretion alone is insufficient for effector identification. Indeed, many secreted proteins might be unrelated to plant parasitism or secreted within the nematode between different cells or organs. A previous analysis using degenerate amino acid classifications identified four protein motifs enriched in the 100 first amino acids of known *M. incognita* effectors [31]. In the *M. incognita* genome, we identified 12,625 proteins harboring at least one of these effector-enriched motifs in their first 100 amino acids (list available at https://doi.org/10.15454/LMY6LV). Cross-referencing this list with that of the 2811 PSP yielded a total set of 2146 PSP bearing an effector-enriched motif and thus representing candidate effectors. Nearly half (1039) of the candidate effectors were specific to PPN, according to OrthoFinder data, which represents a significant enrichment (Pearson’s Chi-squared test with Yates’ continuity correction: χ-squared = 412.3, df = 1, *p*-value < 2.2 × 10^−16^). The list of candidate effectors is available at https://doi.org/10.15454/CSTXU2.

### 3.6. Enriched Functions in PPN-Specific Proteins, PSP and Candidate Effectors Reveal New Promising Targets for RKN Control

InterProScan annotation enabled the identification of at least one known protein domain, motif, or signal in 39,353 *M. incognita* proteins (90% of the predicted proteins in this species). A total of 28,288 of these proteins contained at least one detected InterPro domain, which allowed assignment of at least one GO term to 20,610 *M. incognita* proteins (covering 1939 different GO terms in total). The full InterPro annotation results are available at https://doi.org/10.15454/9BFFKG.

We searched for significantly enriched GO terms in (i) the 12,683 *M. incognita* proteins specific to PPN identified by OrthoFinder, (ii) in the 1331 PPN-specific PSP, and (iii) in the 1039 PPN-specific candidate effectors (Figure 2). Only 9.6% (1222) of the PPN-specific proteins in *M. incognita* had at least one GO term assigned, which were distributed as follows: 1139 terms in the molecular function (MF), 654 in the biological process (BP) and 205 in the cellular component (CC) ontologies. For PPN-specific PSP and candidate effectors, only 7.7% (103) and 7.9% (83) had at least one GO term assigned. This illustrates the high proportion of potential pioneer proteins in these PPN-specific datasets.

In the PPN-specific *M. incognita* proteins, FUNC identified 37, 83 and 45 overrepresented MF, BP, and CC terms, respectively (raw and manually classified FUNC results available online at https://doi.org/10.15454/8NZABA). These terms were further grouped into 16, 7, and 3 over-represented MF, BP, and CC representative terms. Since several protein sets overlapped different GO terms, with some belonging to the same GO sub-graphs, we further classified the enriched terms in the following eight main categories (Figure 2): (1) RNA polyadenylation and Transferase activities; (2) Transcription-related processes and activities; (3) Protein metabolic processes and Peptidase activities; (4) Lyase activity and Extracellular region; (5) Lipid transport and Hydrolase activity; (6) Binding; (7) Ion channel complex; and (8) Signal transduction. We then assessed whether the same categories were also enriched in the PSP and candidate effector subsets of PPN-specific *M. incognita* proteins (Figure 2). We also explored the RNA-seq based transcriptional profiles of genes belonging to each category through the construction and analysis of heatmaps (all data are available at https://doi.org/10.15454/8NZABA/VUKBRU).

Terms related to post-transcriptional modification were strongly enriched in PPN-specific proteins. Surprisingly, 92 out of 99 *M. incognita* proteins annotated under RNA polyadenylation process (GO: 0043631) and 88 out 95 proteins annotated under Polynucleotide adenylyltransferase activity (GO: 0004652) were PPN-specific. These terms were the two most significantly overrepresented terms. Furthermore, several GO terms related to transcriptional processes were also enriched in the PPN-specific proteins, such as DNA-templated transcription, initiation (GO: 0006352); Regulation of transcription; DNA-templated (GO: 0006355); and terms related to DNA binding such as DNA binding function (GO: 0003677); Sequence-specific DNA binding (GO: 0043565), DNA binding transcription factor activity (GO: 0003700); and Zinc ion binding (GO: 0008270). This last GO term was plotted together with other Binding terms (Figure 2). Although all these terms related to transcriptional and post-transcriptional processes were highly enriched in the set of PPN-specific *M. incognita* proteins, none were found to be enriched in the PSP or candidate effector subsets. This indicates these transcription-related proteins are not secreted and probably participate in functions within the nematode cells.

Besides terms related to transcription and post-transcriptional modifications, putative peptidases, which were assigned to the biological processes Proteolysis (GO: 0006508) and Protein deubiquitination (GO: 0016579), both critical post-translational mechanisms, were also enriched in PPN-specific proteins. Additionally, three distinct classes of proteases were enriched: Metalloendopeptidase activity (GO: 0004222), cysteine peptidases (Thiol-dependent ubiquitinyl hydrolase activity (GO: 0036459) and Cysteine-type peptidase activity (GO: 0008234) and threonine peptidase, represented by ATP-dependent peptidase activity (GO: 0004176). While being overrepresented in PPN-specific proteins, most of these peptidase-related GO terms were not overrepresented in PSP or candidate effectors. The only two exceptions were the terms ‘Proteolysis’ and ‘Cysteine-type peptidase activity’, which were overrepresented in PSP but not in the candidate effectors.

Overall, only few GO terms overrepresented in PPN-specific proteins remained overrepresented in PPN-specific candidate effectors. These mainly included terms related to secreted carbohydrate-active enzymes known to be involved in the degradation of the plant cell wall [14], such as Hydrolase activity, hydrolyzing O-glycosyl compounds (GO: 0004553), Pectate lyase activity (GO: 0030570) and Extracellular region (GO: 0005576), as well as Carbohydrate binding (GO: 0030246). In addition to terms related to plant cell wall digestion, two other terms were also found to be enriched in the PPN-specific candidate effectors, namely Chitosanase activity (GO: 0016977) and Neuropeptide signaling pathway (GO: 0007218).

### 3.7. Horizontal Gene Transfers Are More Abundant in PPN-Specific Proteins and Contribute to Some of Their Enriched Functions

Alienness analysis [48] enabled the identification of 1870 *M. incognita* proteins returning an Alien Index (AI) above 0; indicative of a better hit to non-metazoan than metazoan proteins in nr (cf. methods). The whole Alienness results are available at https://doi.org/10.15454/W6SWZH. Of these, 114 returned more than 70% identity to a non-metazoan hit, representing possible contaminations, according to the 70% rule [65]. As such, they were eliminated from further analyses, leaving a total of 1756 proteins possibly acquired via HGT in *M. incognita*.

Nevertheless, an AI > 0 alone only reflects a better hit to non-metazoan than metazoan proteins and is insufficient to assume acquisition via HGT. Higher AI values represent higher amplitudes of difference between the best non-metazoan and metazoan hits. An AI > 14 represents a good balance between HGT prediction accuracy and recall in PPN [48]. Here, Alienness identified 404 *M. incognita* proteins returning an AI > 14, thus likely to have been acquired via HGT from a non-metazoan source.

To estimate the accuracy of our HGT analysis, we compared the proteins identified here with an AI > 14 to the phylogenetically supported HGT cases described in *M. incognita* and other RKN [48] (Table 2).

Of the 15 previously known families of HGT in RKN, 13 could be retrieved with an AI > 14, including all the plant cell wall-degrading enzyme families previously described in RKN [13] and in the PPN-specific section. Two other known HGT families, namely candidate cyanate lyases and expansin-like proteins, returned positive AI values, although below the AI > 14 threshold value. In total, as most form multigene families, the known HGT cases alone represented 124 (>30%) of the 404 proteins with an AI > 14. The rest of the proteins share the same AI properties as the known HGT, and might represent new cases not yet validated by phylogenetic analyses.

To explore the functional impact of genes possibly acquired via HGT, we performed a GO terms enrichment on the set of proteins returning AI scores >0 and >14 (methods, https://doi.org/10.15454/8NZABA and Figure 3). As expected, terms related to the degradation of the plant cell wall, such as Pectate Lyase activity, Polygalacturonase activity, Carbohydrate metabolic process or Hydrolase activity, hydrolyzing O-glycosyl compounds, were enriched both in the possible (AI > 0) and more stringent candidate (AI > 14) HGT sets. Since these cell wall-degrading enzymes are secreted *in planta* by the nematode, the term Extracellular region was also significantly enriched. In addition to these enzymes, other enriched terms in the possible and candidate HGT sets corresponded to previously reported HGT cases, such as Pantothenate biosynthetic process, which corresponds to PanC, a protein probably involved in the biosynthesis of vitamin B5 [66], or Glutamate ammonia ligase activity, corresponding to GSI [67,68], the exact function of which is still unclear in RKN.

Other terms not corresponding to previously known HGT were also enriched in the set of possible HGT (AI > 0). Interestingly, several of these were also enriched in the PPN-specific set presented in the previous section. These included, those related to transcriptional or post-transcriptional processes (DNA binding, DNA-templated transcription, initiation, RNA polyadenalyation), Protein kinase activity, as well as terms related to protein degradation (Proteolysis, Serine-type endopeptidase and ATP-dependent peptidase activities). However, none of these terms were further enriched in the set of candidate HGT (AI > 14), as most of the corresponding proteins returned positive AI values but <14. Whether they actually represent new HGT cases of non-metazoan origin will require further phylogenetic assessment. Actually, only three novel terms (Viral DNA genome packaging, Lyase activity, and Periplasmic space) were enriched in the candidate HGT set. The term ‘Viral DNA genome packaging’ corresponded to a whole set of proteins bearing an InterPro domain named ‘Adenovirus packaging protein 1’, with no evident role in relation to plant parasitism. The enriched terms Lyase activity, and Periplasmic space were associated with proteins bearing an Alginate lyase domain. Alginate is a polysaccharide usually found in brown algae, and whether the RKN possess enzymes that can degrade alginate would be interesting to assess in the future.

Exploring the possible donors of candidate HGT cases showed that 83% (334/404) were of possible bacterial origin, 5% from plants, 4% from fungi, 5% from other non-metazoan eukaryotes, 1.5% from Stramenopiles, while Archaea and Viruses each represented <0.5% of the possible donors. By restricting the analysis to taxonomic information supported by at least three consecutive blast hits, Bacteria represented 94% of the candidate donors, Plants 2%, Fungi 1%; other non-metazoan Eukaryotes 3%, while the other categories (Stramenopiles, Viruses, and Archaea) now disappeared, being supported by a maximum of two consecutive hits.

To estimate the contribution of HGT to the set of PPN-specific genes in *M. incognita*, we cross-referenced the list of PPN-specific proteins with those returning an AI > 0. Only 756 or 14% of the 12,683 PPN-specific proteins in *M. incognita* returned an AI > 0, of which 148 had an AI > 14 and thus are likely to have been acquired via HGT. Hence, only 1% of the PPN-specific proteins have likely been acquired via HGT and possibly up to 14%. Nevertheless, we note that the proportion of possible or candidate HGT in the set of PPN-specific proteins represents a significant enrichment compared to the rest of *M. incognita* proteins (*p*-values of the χ^2^ tests being 2.2 × 10^−16^ and 8.48 × 10^−4^ for AI > 0 and AI > 14, respectively). Moreover, proteins possibly acquired via HGT contribute to some of the enriched functions in PPN-specific proteins, including in candidate effectors, although most candidate effectors have no homology at all and no predicted function.

### 3.8. New Candidate Effectors Amenable to Nematode Control via in Planta Inactivation

One effective strategy towards controlling infection by phytoparasitic nematodes is engineering plants to produce double-stranded RNAs that, in turn, will specifically silence nematode genes essential for parasitism via RNAi-based technologies [69]. A prerequisite is to target nematode genes that are expressed while the parasite is infecting plant tissues, such as genes encoding effectors [17,18,19].

Hence, from the 1039 PPN-specific candidate effector proteins, we selected those encoded by genes with substantial RNA-seq support for expression in endophytic phases of the nematode life cycle. We required the genes to show expression values in these phases above the median (>0.78 and 0.73 for J3-J4 and adult females, respectively). We further required that the average expression in endophytic phases (J3-J4 and adult females) was higher than the average expression during exophytic phases (eggs and pre-parasitic J2).

In order to target genes broadly important for parasitism in RKN species, we only retained proteins conserved in *M. incognita* and at least two more RKN species. Being specific to PPN yet conserved in multiple RKN species, these candidate effectors are more likely to play core parasitic functions in these species.

In addition, to facilitate downstream biological validations and assays, we eliminated candidates that originated from multigene families. Indeed, designing specific probes for targeting one single gene in a multigene family can be challenging. Furthermore, the monitoring of silencing effects would be more difficult, with possible compensation of the loss of activity/function. Given that the *M. incognita* genome is allo-triploid, many genes are present in three homoeologous copies (i.e., orthologs that have been brought back together in the same genome via inter-species hybridization) [23]. Consequently, all the proteins present in more than three copies were considered as potentially originating from a multigene family and were therefore eliminated.

The application of all the above criteria led to the selection of 89 candidate effector proteins in *M. incognita* that were specific to PPN species, according to OrthoFinder (list available at https://doi.org/10.15454/WBZZ5M). 

Of these 89 *M. incognita* candidate effectors, only 21 returned hits against the NCBI’s nr, of which 13 corresponded to previously published *M. incognita* effectors and one to a putative *M. javanica* effector (Diamond hit accession numbers in bold in https://doi.org/10.15454/WBZZ5M). Since we aimed at identifying novel candidate effectors, lacking homology in other species, the 21 *M. incognita* proteins returning hits were eliminated.

The 68 remaining proteins returned no hits against the NCBI’s nr library and represent novel PPN-specific candidate effectors.

### 3.9. Expression Validation of Candidate Effector Genes

Many important effectors are developmentally regulated, with an increase in expression during endophytic phases of the RKN life cycle [70,71]. From the 68 novel candidate effectors identified in this study, and expressed in endophytic stages, according to in silico RNA-seq data, we randomly selected seven genes (Table 1) for independent validation of expression pattern in *M. incognita* during parasitism on *N. tabacum*. In addition, a previously identified candidate *M. incognita* effector, meeting the same selection criteria, Minc3s00292g09561 (= Minc02654) was included as a positive control in expression analysis. This gene was previously described as overexpressed during *M. incognita* parasitic stages [72].

Using RT-qPCR analysis, we monitored gene expression in eggs, pre-parasitic J2 (ppJ2), and endophytic *M. incognita* stages at 5, 10, 15, and 22 days after nematode inoculation (DAI). Expression modulation was compared with the in silico RNA-seq whole transcriptome data for nematode life-cycle stages described in the previous section (Egg, ppJ2, pJ2-J3-J4 and Female). Overall, all seven candidate genes displayed significantly increased expression in at least one of the endophytic phases of the life cycle (5, 10, 15 or 22 DAI), in comparison to their expression in the egg and pre-parasitic J2 stages (Figure 4). These expression trends are consistent with the RNA-seq-derived expression patterns. Furthermore, RT-qPCR analysis of the positive control gene (Figure 4H) also showed expression modulation patterns matching those previously described [72].

Similarly to this previously described effector, the effector candidates Minc3s03880g3504, Minc3s00020g1309 and Minc3s01206g21700 displayed a sharp increase in their expression during the juvenile endophytic stages pJ2-J3-J4, and continued to be expressed in the adult female stage, albeit at a lower level for Minc3s03880g3504 (Figure 4E–G). Therefore, these genes constitute interesting targets for *in planta* delivered RNAi strategy.

In addition to the parasitic J2 stage, *M. incognita* nematodes have relatively short and regular feeding cycles at adult female stages [73]. We found four effector candidates, Minc3s00016g00668, Minc3s01635g25253, Minc3s00010g00666 and Minc3s1217g21793, which returned the highest expression in females (22 DAI), with low to no expression at earlier stages (Figure 4A–D). These proteins are probably involved in feeding site maintenance, and therefore also constitute potential targets for RNAi.

## 4. Discussion

### 4.1. PPN-Specific Proteins Reveal Novel Candidate Effectors and Other Promising Functions for Nematode Control

Using an extensive comparative genomics analysis, we identified a whole set of novel candidate effectors in *M. incognita* that have no recognizable homologs in species other than PPN. As they occur in several *Meloidogyne* or other PPN from H-clade 12, these genes probably play important roles in plant parasitism. Furthermore, as they are exclusive to PPN, the specific targeting of these genes would reduce the risk of unintentional off-target effects on other species. Finally, as they are expressed during endophytic phases of the nematode parasitic life cycle, these genes are amenable to in planta RNAi silencing and thus constitute a promising set of targets for the development of cleaner and more specific strategies to control nematode pests.

Although we focused on candidate effectors, representing the most obvious targets, functional analysis of the ensemble of PPN-specific proteins revealed other novel and interesting candidates. As expected, because the majority of these PPN-specific proteins lack recognizable homology, only a few possess a predicted conserved domain to allow functional annotation. Indeed, only 9.6% of the PPN-specific proteins in *M. incognita* had at least one GO term assigned, in comparison to more than 47% for the whole proteome. This illustrates the high proportion of potential pioneer proteins without evident functional annotation in the PPN-specific datasets.

Despite this, among the few PPN-specific proteins with functional annotation, some terms were significantly enriched, with functions deserving further attention in the future. For instance, a series of GO terms related to transcriptional and post-transcriptional activities were enriched in PPN-specific proteins, but not further enriched in the secreted or candidate effector subsets, indicating an internal function in the plant-parasitic nematode. Interestingly, these terms were also enriched in the possible HGT protein set (AI > 0), indicating higher similarity to non-metazoan than metazoan proteins. Whether they represent possible acquisition via horizontal gene transfer will need further phylogenetic investigation. Regardless of their origin, these proteins might be involved in the regulation of PPN-specific functions, including control of effector gene expression or development of organs directly involved in parasitism.

Another interesting term, ‘Neuropeptide signaling pathway’ was enriched in PPN-specific proteins and associated with four proteins bearing a signal peptide for secretion, no transmembrane region and a ‘FMRFamide related peptide family’ domain. FMRFamide-like peptides (FLPs) are widely conserved neuropeptides in nematodes that likely play a variety of roles from locomotion to feeding [74]. These *M. incognita* FLP proteins have no predicted homologs in nematodes other than plant parasites and might therefore be involved in phytoparasitic-specific functions, such as sensing, recognition of the host plant, and attraction towards the roots.

Finally, terms related to metallo and cysteine peptidase activity were also enriched in the set of *M. incognita* PPN-specific proteins. In nematodes, peptidases of these families are involved in molting and ecdysis processes, as a new cuticle is synthesized five times during nematode development [75]. The metalloendopeptidase genes we identified in *M. incognita* are mainly expressed in eggs and in pre-parasitic J2 larvae while the cysteine-type peptidases are highly expressed in late parasitic forms and females. As in the RKN and other PPN, several molts take place within the plant tissue, with these peptidases probably also involved in PPN-specific molts or cuticle modifications. Preventing nematode molting in the soil or within plant tissues would lead to developmental arrest and failure of parasitism.

Overall, the present comparative genomics analysis not only revealed new and interesting candidate effectors, but also identified other families of proteins that improve our understanding of the arsenal of genes employed by nematodes to manipulate their hosts or that are essential for development and survival. Now being publicly available, these data constitute a valuable resource to fuel future research in the field of plant—nematode interactions.

Reinforcing their potential importance, certain PPN-specific candidate effectors seem to show similar expression patterns in different *M. incognita* isolates and host plants. Indeed, from the selected candidate effectors more highly expressed in endophytic stages of the parasitic life cycle, according to previously generated RNA-seq data [23], RT-qPCR assays on seven genes confirmed higher expression during endophytic phases too. The RNA-seq assay was performed with an *M. incognita* strain originally from Mexico (strain Moreols) on a susceptible *S. lycopersicum* host (tomato, var. *esculentum* cv. St Pierre), whereas the RT-qPCR assays were conducted with an *M. incognita* strain from Brazil as well as from a different compatible host plant (tobacco). This consistency in expression suggests the examined genes are broadly important for *M. incognita* infection on different plant hosts and strengthen their potential as targets for the development of efficient nematode control methods.

### 4.2. Evolutionary Origin of PPN-Specific Genes

Comparative analysis of 61 nematode genomes revealed 12,683 *M. incognita* proteins with no orthologs in nematodes other than PPN. One can thus wonder about the origin of these numerous proteins, which appear to be absent from the other nematodes. These proteins are unlikely to result from overpredictions by gene calling software, as the vast majority are supported by substantial expression data and are conserved in multiple PPN genomes. Further homology searches against the NCBI’s nr and Alienness analysis showed HGT to have likely contributed to 1% (AI > 14) of the 12,683 PPN-specific genes in *M. incognita,* and up to 14% if we consider possible HGT (AI > 0). Some of the functions enriched in the PPN-specific genes, including those related to degradation of the plant cell wall or transcription, are clearly due to genes possibly acquired by HGT. Candidate HGT also revealed new proteins not previously described in PPN, such as ‘adenovirus packaging protein 1’ and ‘alginate lyase’. The origin and possible function of these proteins in Meloidogyne and other PPN deserve further investigation.

Although HGT are significantly enriched in the PPN-specific dataset, this only explains the origin for a minority of the genes (maximum 14%). This is consistent with the observation that 88% of the PPN-specific *M. incognita* proteins return no significant hit at all against the NCBI’s nr library. Thus, these proteins are more likely encoded by lineage-specific orphan genes, with two main hypotheses to explain their evolutionary origin [76]. The first postulates that duplications followed by rearrangements and extremely rapid sequence divergence can lead to gene copies lacking recognizable homology to other sequences. A second hypothesis, of *de novo* gene birth, had long been overlooked, but is now gaining greater attention. Under this hypothesis, protein-coding genes can emerge *de novo* from non-genic regions by two main mechanisms: one called RNA first and the other called ORF first [77]. RNA first postulates that an ORF can emerge by accumulation of mutations in an originally transcribed non-coding region. ORF first, in contrast, postulates that a coding but untranscribed region can acquire cis-mutations (e.g., insertion of a transposable element) and become transcriptionally active. As these PPN-specific proteins represent a substantial part of the proteome and include known and new candidate effectors, their contribution both to the genome and biology of the RKN is important. These findings are consistent with previous reports indicating that most of the effectors are ‘orphan’ proteins, lacking homology outside the PPN [11]. Further investigation into the evolutionary origin of these PPN-specific genes and of their actual impact on nematode biology is recommended. Available genomes from close outgroups of the Tylenchida PPN will be a prerequisite to search for similarities to the PPN-specific genes in these species. Hopefully, such data can be expected in the near future, enabling investigations of their ancientness and whether they emerged from pre-existing non-genic regions.

### 4.3. An Unprecedented Representation of PPN in a Comprehensive and Phylogenetically Accurate Comparative Framework

Although comparative phylogenomics analyses of nematodes have previously been published, these studies only encompassed two [32] or three [39] genomes for PPN species, limiting insight in genome evolution associated with plant parasitism. In the present study, we conducted a phylogenomics analysis that covered a diversity of nematode clades and lifestyles and included 16 PPN species. This phylogenomics comparison offers an unprecedented representation of phytoparasitic nematodes and constitutes a new opportunity to study genomic singularities at the protein family level, associated with the evolution of plant parasitism in nematodes.

However, comparative analysis can only express its full potential if supported by an underlying phylogenetically correct framework. We thus compared the phylogeny of the species obtained by our phylogenomics analysis to the reference nematode classification systems.

Two main classification systems have been developed to describe the nematode diversity. The system described by Mark Blaxter and colleagues [58,59] defined five clades (I–V) that we have referred to as B-clades. This system is based on a limited number of species but employs concatenated multiple marker genes. In parallel, the classification system described by Hans Helder and colleagues defined 12 clades [1,60] and, although it comprises the largest diversity of nematodes to date (including PPN); it is based on only one single marker gene (SSU rRNA). We have referred to this classification system as H-clades (1–12). The backbones of these two classification systems have recently been reinforced and further confirmed by a phylogenomics analysis based on a concatenated multigene alignment comprising 108 nematode species [62], representing a good balance between the phylogenetic information in the concatenated markers and species diversity.

The phylogenetic species tree we obtained by OrthoFinder analysis was remarkably consistent with the recently published phylogenomics study of nematodes [62], and by extension, with the two historical classification systems in B and H clades. Hence, the comparative analysis of the protein sets, including at the level of clade-specific proteins and proteins shared by multiple species, is supported by a phylogenetically correct background, allowing evolutionarily relevant comparisons. Not only was the OthoFinder topology almost identical to that of [62]; but the support values for the different clades were all >0.9 (except in one branch), indicating a strong signal supporting the phylogeny. The only support values lower than 0.9 concerned the relative branching order of the RKN within clade I, *sensu* De Ley (*M. arenaria*, *M. enterolobii*, *M. floridensis*, *M. incognita* and *M. javanica*). This observation is not particularly surprising, as these RKN species underwent complex hybridization and polyploidization events, with a probable same common or very similar maternal origin and distinct paternal progenitors [23,24,29,78]. As a result, some genome regions are highly similar between these species, while copies of the region are highly divergent within a same species. These complex homology relationships complicate both one-to-many and many-to-many orthogroup analyses in these allopolyploid species. Nonetheless, because the gene copies are more similar to each other within RKN clade I than to homologs in the other RKN clades, most of the orthogroup trees will show a monophyletic group encompassing all these allopolyploid nematode species. This explains the high support for the monophyly of clade I in Figure 1. However, depending on the gene and homoeolog gene copy under consideration in a given orthogroup, the relative positions of the corresponding proteins between the four allopolyploid RKN species will vary and lead to lower support values. This lower support for relative positions of species among clade I RKN have no influence on our comparative analysis, as we mainly focused on *M. incognita* genes specific to PPN and did not draw conclusions on phylogenomic differences between the four clade I RKN.

## 5. Conclusions

Using comparative genomics analysis on a broad sample of nematode species selected for their diversity of phylogenetic position and lifestyles, as well as the quality of their genomes, we have identified a whole set of proteins specific to plant-parasitic species. Focusing more specifically on the model and devastating root-knot nematode *M. incognita*, we found that ca. 10,000 genes (a quarter of the predicted proteins) are only found in other plant-parasitic nematodes. A small proportion of these proteins show similarity with non-metazoan species and might originate from HGT. However, the vast majority returned no significant similarity at all, and their origin remains, so far, elusive. Since most of these proteins are conserved in multiple PPN species and the corresponding genes are supported by expression data, they are unlikely to result from in silico over-predictions. Although *de novo* gene birth is a process that has recently attracted greater attention, elucidation of the origin of these PPN-specific genes will need further and more focused investigation. Regardless of their origin, some functions enriched in these PPN-specific genes point towards new processes such as neuro-signaling, molting, or transcription, that have not yet been investigated as targets for development of control strategies of phytoparasitic nematodes.

Targeting PPN-specific genes important for parasitism appears as a promising prospect in the quest towards controlling these devastating plant pests while minimizing the negative environmental impact. Within this perspective, candidate effector proteins constitute obvious target genes, given their importance in parasitism. Here, we identified ca. 1000 PPN-specific proteins in *M. incognita* sharing the same characteristics as known effectors. Targeting specific genes of an endoparasite can be conducted through plant transformation and RNA-mediated gene silencing. Towards this, we further filtered the dataset to retain a number of novel PPN-specific candidate effectors expressed while the nematode is within the plant. Employing RT-qPCR, we confirmed their higher expression during endophytic phases of parasitism on a second host species, suggesting a conserved and important role in plant parasitism.

Collectively, these datasets of proteins constitute promising resources in contributing to a better understanding of the genetic basis of nematode adaptation to phytoparasitism and future development of more efficient and specific nematode control strategies.

## Figures and Tables

**Figure 1 genes-11-01347-f001:**
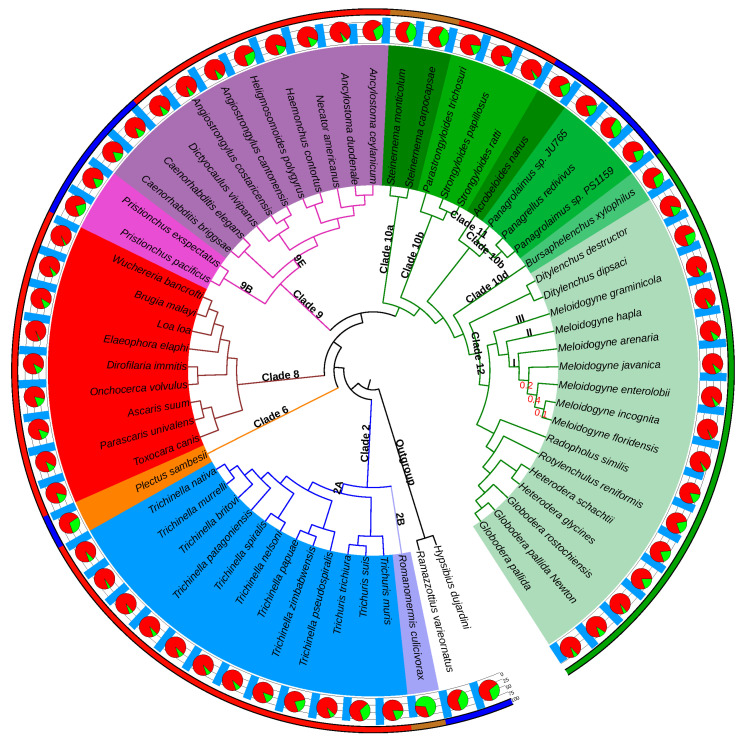
Annotated phylogenomic tree of the compared nematode genomes. Topology of the phylogenetic tree was obtained from the OrthoFinder comparative analysis of 61 nematode genomes and two outgroup tardigrade genomes. Latin species names are given at the leaves. STRIDE support values <0.9 are provided at the corresponding branches in red (all the other values are ≥0.9). Nematode clade numbers, according to the Helder classification (H-clades), are provided at the corresponding branches. Roman numbers within clade 12 correspond to the root-knot nematodes (RKN) clades, according to De Ley [63]. For each species, the pie-charts represent the relative proportion of species-specific proteins (green) and proteins conserved in at least another species (red). The light blue bar histogram represents the percentage of complete eukaryotic BUSCO proteins in each proteome. The color-coded outermost circle is according to the species lifestyle (blue: free-living, brown: insect parasite, green: plant parasite, red: vertebrate parasite). This figure was generated with Itol [64].

**Figure 2 genes-11-01347-f002:**
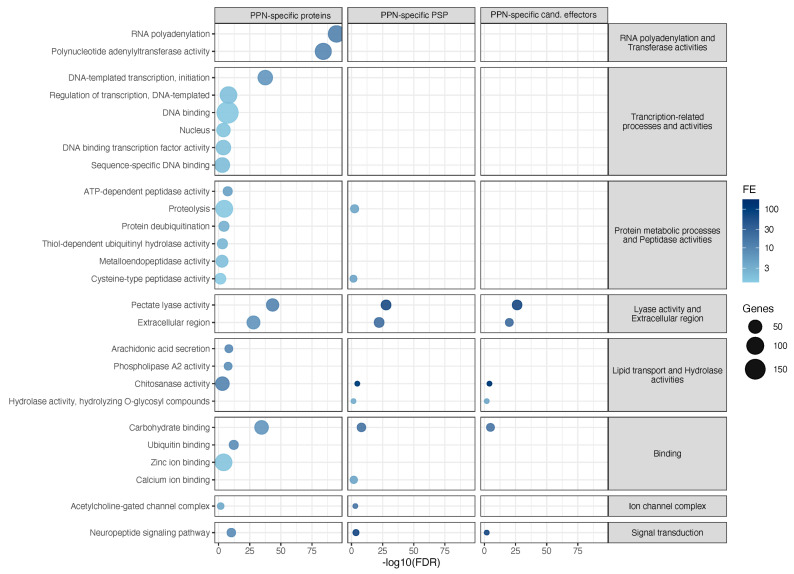
Enriched GO terms in plant-parasitic-specific proteins, including putative secreted proteins and candidate effectors. Only GO terms with FDR <0.05 and at least four genes in the node are represented. Correlated terms are grouped together and represented by more general terms (right-side grey rectangles). Bubble sizes are according to the number of genes annotated with a specific GO term. Bubble positions are according to −log10 (FDR) (false discovery rate) values. The darkness level of bubble colors is according to GO terms fold-enrichment (FE) values (see methods).

**Figure 3 genes-11-01347-f003:**
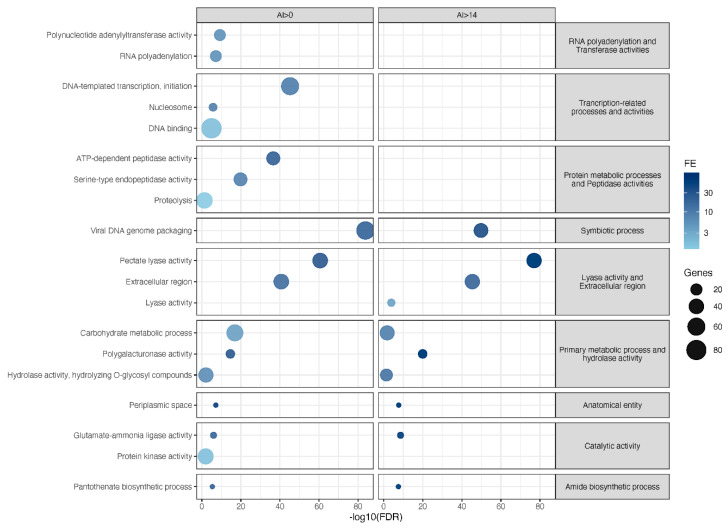
Enriched GO terms in proteins possibly acquired via horizontal transfers. Left side: possible HGT—AI > 0; right side: likely HGT—AI > 14. Only GO terms with FDR < 0.05 and at least four genes in the node are represented. Correlated terms are grouped together and represented by more general terms (right-side grey rectangles). Bubble sizes are according to the number of genes annotated with a specific GO term. Bubble positions are according to −log10 (FDR) values. The darkness of bubble colors is according to GO terms fold-enrichment (FE) values (see methods).

**Figure 4 genes-11-01347-f004:**
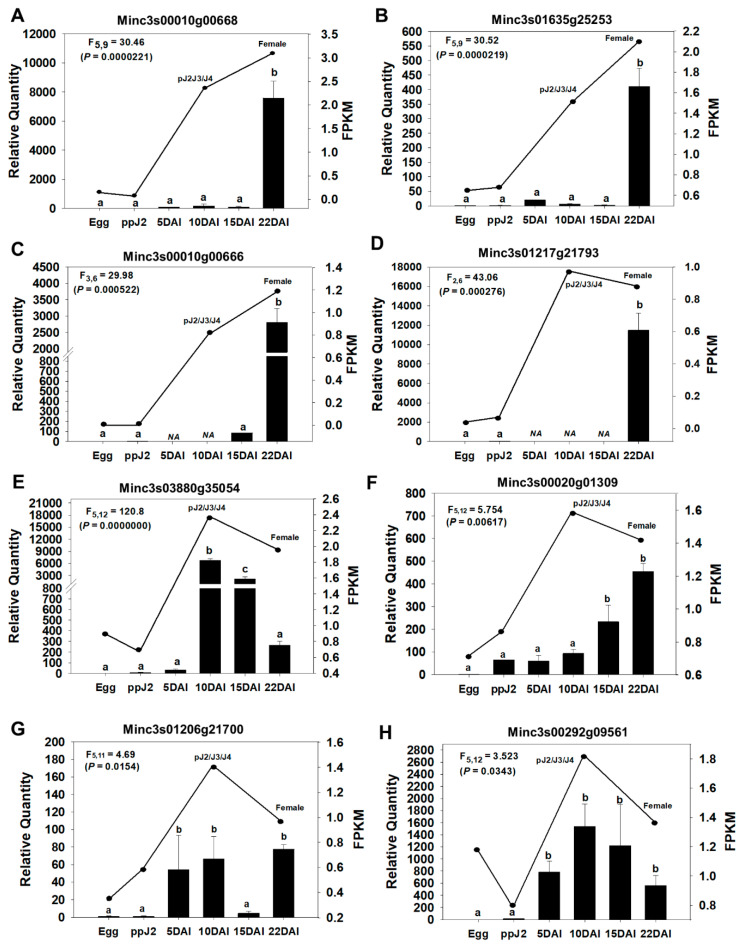
RNA-seq and RT-qPCR expression profiles of PPN-specific candidate effector genes during the *M. incognita* life cycle. Relative mRNA levels of eight genes (**A**–**H**) at different stages of the nematode life cycle during its compatible interaction with *Nicotiana tabacum.* RT-qPCR data are shown in bars (mean ± standard error) with values in the left “y” axis and analyzed in six stages: egg, ppJ2, and endophytic phases at 5, 10, 15, and 22 days after inoculation (DAI). RNA-seq in silico data are shown as a line plot with values displayed at the right “y” axis (FPKM) and analyzed at four stages: egg, ppJ2, and endophytic phases pJ2/J3/J4 and female. RT-qPCR expression data were analyzed using the SATqPCR tool [57], with expression values statistically analyzed using ANOVA and Tukey’s test (*p* < 0.05). Different letters indicate statistically significant differences between samples.

**Table 1 genes-11-01347-t001:** Primers used for RT–qPCR analysis.

Candidate Effector	Primer Sequence (5′–3′) Forward/Reverse	Amplicon Size (bp)	Primer Efficiency
Minc3s00010g00668	CTTGGTACCCTTACCGACCA/ AAGTCGCACTTGTTCTGGTG	110	0.91
Minc3s01635g25253	TGCAAAGATGGGTGTACAAT/ TTTGCACATCGCTTTCTCAC	113	0.94
Minc3s00010g00666	ACCGATTCGTTCAGTTCCAG/ CCTCATTATCCATCGGTGCT	89	0.91
Minc3s01217g21793	CAGGACGTTCGGTTCCAATA/ TGCTGTGGCATGACGTTTAG	122	0.89
Minc3s03880g35054	CGAAATGGGCGTAGAAAATG/ GTCGGCCATGTGGTACTTCT	83	0.88
Minc3s00020g01309	CCCAAAGCAATGCAACATAA/ ACGAATGTGCCGAAGAGAAT	97	0.87
Minc3s01206g21700	TCCAAATTGCGTGGTAGACA/ GCTTGTGAAATGCGTCAGAA	104	0.88
Minc3s00292g09561	AAGAGGAGTGTGGGGTTGTG/ GCTTGGAAGAATTGGACGAC	94	0.89

**Table 2 genes-11-01347-t002:** Phylogenetically confirmed horizontal gene transfer (HGT) cases in RKN retrieved by Alienness.

Confirmed HGT	Representative InterPro Domain	Highest AI	Candidates with AI > 14	Process ^1^
GH5_2 Cellulases	Glycoside hydrolase, family 5 (IPR001547)	32.57	21	PCW degradation
PL3 Pectate Lyase	Pectate lyase PlyH/PlyE-like (IPR004898)	121.94	39	PCW degradation
GH30 xylanase	Glycoside hydrolase family 30 (IPR001139)	191.63	19	PCW degradation
GH28 Polygalacturonase	Glycoside hydrolase, family 28 (IPR000743)	302.29	15	PCW degradation
GH43 Candidate Arabinanase	Glycoside hydrolase, family 43 (IPR006710)	174.53	3	PCW degradation
Expansin-like proteins	RlpA-like protein, double-psi β-barrel domain (IPR009009)	8.54	0	PCW degradation
Candidate Isochorismatase	Isochorismatase-like (IPR000868)	51.56	3	Def. Manipulation
Chorismate mutase	Chorismate mutase II, prokaryotic-type (IPR002701)	30.85	4	Def. Manipulation
GH32 invertase	Glycoside hydrolase, family 32 (IPR001362)	306.79	6	Nutrient processing
VB5 PanC	Pantoate-β-alanine ligase (IPR003721)	137.24	2	Nutrient processing
Candidate GSI Glutamine Synthase	Candidate GSI Glutamine Synthase (IPR008146)	248.7	5	Nutrient processing
NodL-like	Hexapeptide transferase, conserved site (IPR018357)	136.85	2	Feed. site induction
Candidate L-threonine Aldolase	Aromatic amino acid β-eliminating lyase/threonine aldolase (IPR001597)	271.61	3	Unknown
Candidate Phosphoribosyltransferase	Phosphoribosyltransferase domain (IPR000836)	154.59	2	Unknown
Candidate Cyanate Lyase	Cyanate lyase, C-terminal (IPR003712)	3.49	0	Detoxification

^1^ PCW: plant cell wall; Def. manipulation: manipulation of plant defense; Feed. Site induction: induction of the nematode feeding site from plant giant cells.

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
