# Peer review of "Comparative Genomics Reveals Novel Target Genes towards Specific Control of Plant-Parasitic Nematodes"

_genes, 2020, doi:10.3390/genes11111347_

Round 1

Reviewer 1 Report

This is an impressive and comprehensive body of work. The thoroughness of the multiple approaches taken herein give very strong confidence in the results presented, and the authors rightly point out that this will be an invaluable resource for scientists studying plant parasitism by nematodes. I have no specific comments on the work, as the approach was logical, used appropriate and diverse analysis tools, and data were well interpreted and described. I am particularly impressed by the 61 species phylogenetic analysis. Not only is this an impressive stand-alone achievement, but the correlation to previously published work is an important contribution. My only issue is the occasional odd usage of English. For example, in L18 I think the authors mean to say "extensive", not "expressive". L682; not sure what "reinforcing their interest" means. These types of very minor issues occur throughout, and I would recommend proof reading by an english editor.

Other than those minor bits, this is a remarkable piece of work and a well written and prepared manuscript. 

Author Response

This is an impressive and comprehensive body of work. The thoroughness of the multiple approaches taken herein give very strong confidence in the results presented, and the authors rightly point out that this will be an invaluable resource for scientists studying plant parasitism by nematodes. I have no specific comments on the work, as the approach was logical, used appropriate and diverse analysis tools, and data were well interpreted and described. I am particularly impressed by the 61 species phylogenetic analysis. Not only is this an impressive stand-alone achievement, but the correlation to previously published work is an important contribution.

We would like to thank reviewer 1 for the very positive and encouraging comments and are glad the work presented in this article was appreciated.

My only issue is the occasional odd usage of English.

For example, in L18 I think the authors mean to say "extensive", not "expressive".

L682; not sure what "reinforcing their interest" means.

These types of very minor issues occur throughout, and I would recommend proof reading by an english editor.

We have corrected these points, as well as the English language and punctuation at lines 18, 20, 21, 29, 30, 33, 34, 54, 56, 58, 62, 74, 75, 96, 99, 106, 108, 110-112, 114, 120, 124, 131, 132, 134, 165, 190, 2020, 203, 223, 227, 284, 304, 305, 319, 323, 341, 350-352, 381, 386, 391, 421, 423, 432, 444, 464, 519, 531, 539, 557, 572, 577, 582, 586-587, 606, 610, 619, 626, 630, 634, 636, 637, 643, 653, 657, 672, 679, 680, 686, 688, 689, 692, 693, 695, 703, 788 and 804

Other than those minor bits, this is a remarkable piece of work and a well written and prepared manuscript.

Reviewer 2 Report

This manuscript is very well designed and executed. The author detail a comparative analysis of nematode proteins, extensively annotate these proteins, and test their efficacy for reducing nematode reproduction.  That being said, there are a few minor problems that can be addressed to improve the clarity of the manuscript.

Line 81:  the authors might consider that most effectors not being conserved, is likely due to the majority of them not being identified in most nematode species

Lines 88-110:  The organization is weird in this section and clarity needs to be improved.

Lines 97-99:  This sentence is poorly explained and will be confusing to the reader.

Line 104: paragraph missing topic sentence.

Line 108: needs citation

Lines 119-122: is unnecessary and can be deleted.

Line 412:  This value seems strange to me.  12k of the proteins harbor this motif, almost a third of the genome? It should be made clear that the proteins that were observed, were not just different isoforms from the same gene. Were only primary isoforms used?

Line 461:  What is “the last one”?

Line 465:  I completely agree with this conclusion!

Line 467:  Needs something to transition from the previous paragraph, because I read it and wondered why we were suddenly talking about peptidases. 

Line 559-560:  This is a confusing transition to the next paragraph.

Line 573-574:  Perhaps add some more explanation on here as to the extent of desired “broad effect”.  The introduction spends a fair bit of time explaining why something specific to individual species is desired, why do we want a broad effect here?

Line 579:  the use of “copies”  here needs to be defined.

Line 682-691:  This seems a little speculative as the sample size is pretty small here.

References:  bold faced years are not consistent among the references.

Author Response

This manuscript is very well designed and executed. The author detail a comparative analysis of nematode proteins, extensively annotate these proteins, and test their efficacy for reducing nematode reproduction.  That being said, there are a few minor problems that can be addressed to improve the clarity of the manuscript.

We would like to thank Reviewer 2 for the positive and encouraging comments as well as for the suggestions, which helped improving the clarity of the manuscript. Below is a point by point response to the different points raised.

Line 81:  the authors might consider that most effectors not being conserved, is likely due to the majority of them not being identified in most nematode species

We agree that the lack of overlap between the sets of known effectors of root-knot nematodes and cyst nematodes could simply be due to the fact many are actually not yet identified as effectors and thus more overlap could be expected with progress in effector identification in the different nematode species. We have now added this idea in the manuscript.

Lines 88-110:  The organization is weird in this section and clarity needs to be improved.

We have rewritten this section and hope this has improved the clarity.

Lines 97-99:  This sentence is poorly explained and will be confusing to the reader.

We have now better explained the problem with signal peptide predictions and the fact that some known effector proteins are actually secreted but do not bear a recognizable signal peptide.

Line 104: paragraph missing topic sentence.

We have revised the entire section (lines 88-110), including this paragraph.

Line 108: needs citation

This is citation [32], we have added a new citation to this reference, here.

Lines 119-122: is unnecessary and can be deleted.

We deleted the second sentence but kept the first as we think it is important to mention that the previous comparative genomics analysis only included two genomes for plant-parasitic nematodes. This constitutes a good transition with the next paragraph which explains that substantial progress in nematode genomics has been achieved in between.

Line 412:  This value seems strange to me.  12k of the proteins harbor this motif, almost a third of the genome? It should be made clear that the proteins that were observed, were not just different isoforms from the same gene. Were only primary isoforms used?

We understand this result could seem surprising. These motifs have been identified as significantly enriched in the 100 first amino acids of known RKN effector proteins as compared to evolutionarily widely conserved housekeeping proteins (i.e. presumably not effectors) in these species (Vens et al, Bioinformatics, 2011). Although these motifs are enriched in known effectors, many other proteins also bear the same motifs, and this does not mean that all of these proteins actually are effectors. Presence of this motif alone is not sufficient to identify candidate effectors but the conjunction of these motifs plus the presence of a signal peptide for secretion and no detection of a transmembrane region is required.

This 12k number does not correspond to isoforms from the same genes but to different genes as no isoforms were predicted in the M. incognita genome used in this analysis (Blanc-Mathieu et al., PLoS Genetics, 2017). There is only one isoform per gene in this genome annotation.

Line 461:  What is “the last one”?

We refer here to the Zinc ion binding (GO:0008270) term, this has been clarified in the revised manuscript.

Line 465:  I completely agree with this conclusion!

We are glad reviewer 2 adheres to this conclusion.

Line 467:  Needs something to transition from the previous paragraph, because I read it and wondered why we were suddenly talking about peptidases. 

We agree and have now added a transition sentence to explain these are other over-represented GO terms within PPN-specific proteins, besides the ones related to transcription.

Line 559-560:  This is a confusing transition to the next paragraph.

We now made clear that most candidate effectors present no homology and have no predicted function at all. This should constitute a better transition to the next paragraph.

Line 573-574:  Perhaps add some more explanation on here as to the extent of desired “broad effect”.  The introduction spends a fair bit of time explaining why something specific to individual species is desired, why do we want a broad effect here?

In the whole manuscript, we have described PPN-specific proteins and here we refer to M. incognita proteins specific to PPN, these proteins are not necessarily species-specific and can be conserved in multiple different PPN species. The idea here is to only retain PPN-specific proteins to minimize the risk of effects on other ‘non-target’ species. Then, among the PPN-specific M. incognita proteins, we selected those conserved in multiple different RKN species. Being specific to PPN yet conserved in multiple RKN species, these candidate effectors are more likely to play core parasitic functions in these species. We have clarified this point in the revised manuscript.

Line 579:  the use of “copies”  here needs to be defined.

Homoeologous gene copies refer to orthologs that have been brought together inside a same species as a result of inter-species hybridization. Being allo-triploid, M. incognita has many genes present in three homoeologous copies. We have now clarified the definition.

Line 682-691:  This seems a little speculative as the sample size is pretty small here.

We fully agree and have changed this paragraph to clarify that here we are only referring to the seven genes that have been tested by RT-qPCR.

References:  bold faced years are not consistent among the references.

This manuscript is very well designed and executed. The author detail a comparative analysis of nematode proteins, extensively annotate these proteins, and test their efficacy for reducing nematode reproduction.  That being said, there are a few minor problems that can be addressed to improve the clarity of the manuscript.

We would like to thank Reviewer 2 for the positive and encouraging comments as well as for the suggestions, which helped improving the clarity of the manuscript. Below is a point by point response to the different points raised.

Line 81:  the authors might consider that most effectors not being conserved, is likely due to the majority of them not being identified in most nematode species

We agree that the lack of overlap between the sets of known effectors of root-knot nematodes and cyst nematodes could simply be due to the fact many are actually not yet identified as effectors and thus more overlap could be expected with progress in effector identification in the different nematode species. We have now added this idea in the manuscript.

Lines 88-110:  The organization is weird in this section and clarity needs to be improved.

We have rewritten this section and hope this has improved the clarity.

Lines 97-99:  This sentence is poorly explained and will be confusing to the reader.

We have now better explained the problem with signal peptide predictions and the fact that some known effector proteins are actually secreted but do not bear a recognizable signal peptide.

Line 104: paragraph missing topic sentence.

We have revised the entire section (lines 88-110), including this paragraph.

Line 108: needs citation

This is citation [32], we have added a new citation to this reference.

Lines 119-122: is unnecessary and can be deleted.

We deleted the second sentence but kept the first as we think it is important to mention that the previous comparative genomics analysis only included two genomes for plant-parasitic nematodes. This constitutes a good transition with the next paragraph which explains that substantial progress in nematode genomics has been achieved in between.

Line 412:  This value seems strange to me.  12k of the proteins harbor this motif, almost a third of the genome? It should be made clear that the proteins that were observed, were not just different isoforms from the same gene. Were only primary isoforms used?

We understand this result could seem surprising. These motifs have been identified as significantly enriched in the 100 first amino acids of known RKN effector proteins as compared to evolutionarily widely conserved housekeeping proteins (i.e. presumably not effectors) in these species (Vens et al, Bioinformatics, 2011). Although these motifs are enriched in known effectors, many other proteins also bear the same motifs, and this does not mean that all of these proteins actually are effectors. Presence of this motif alone is not sufficient to identify candidate effectors but the conjunction of these motifs plus the presence of a signal peptide for secretion and no detection of a transmembrane region is required.

This 12k number does not correspond to isoforms from the same genes but to different genes as no isoforms were predicted in the M. incognita genome used in this analysis (Blanc-Mathieu et al., PLoS Genetics, 2017). There is only one isoform per gene in this genome annotation.

Line 461:  What is “the last one”?

We refer here to the Zinc ion binding (GO:0008270) term, this has been clarified in the revised manuscript.

Line 465:  I completely agree with this conclusion!

We are glad reviewer 2 adheres to this conclusion.

Line 467:  Needs something to transition from the previous paragraph, because I read it and wondered why we were suddenly talking about peptidases. 

We agree and have now added a transition sentence to explain these are other over-represented GO terms within PPN-specific proteins, besides the ones related to transcription.

Line 559-560:  This is a confusing transition to the next paragraph.

We now made clear that most candidate effectors present no homology and have no predicted function at all. This should constitute a better transition to the next paragraph.

Line 573-574:  Perhaps add some more explanation on here as to the extent of desired “broad effect”.  The introduction spends a fair bit of time explaining why something specific to individual species is desired, why do we want a broad effect here?

In the whole manuscript, we have described PPN-specific proteins and here we refer to M. incognita proteins specific to PPN, these proteins are not necessarily species-specific and can be conserved in multiple different PPN species. The idea here is to only retain PPN-specific proteins to minimize the risk of effects on other ‘non-target’ species. Then, among the PPN-specific M. incognita proteins, we selected those conserved in multiple different RKN species. Being specific to PPN yet conserved in multiple RKN species, these candidate effectors are more likely to play core parasitic functions in these species. We have clarified this point in the revised manuscript.

Line 579:  the use of “copies”  here needs to be defined.

Homoeologous gene copies refer to orthologs that have been brought together inside a same species as a result of inter-species hybridization. Being allo-triploid, M. incognita has many genes present in three homoeologous copies. We have now clarified the definition.

Line 682-691:  This seems a little speculative as the sample size is pretty small here.

We fully agree and have changed this paragraph to clarify that here we are only referring to the seven genes that have been tested by RT-qPCR.

References:  bold faced years are not consistent among the references.

Bold faced years were missing in all references relative to book chapters, we have now consistently put all years bold faced.Bold faced years were missing in all references relative to book chapters, we have now consistently put all years bold faced.